# The Amyloid-Tau-Neuroinflammation Axis in the Context of Cerebral Amyloid Angiopathy

**DOI:** 10.3390/ijms20246319

**Published:** 2019-12-14

**Authors:** Pablo Cisternas, Xavier Taylor, Cristian A. Lasagna-Reeves

**Affiliations:** 1Stark Neurosciences Research Institute, Indiana University School of Medicine, Indianapolis, IN 46202, USA; pcistern@iu.edu (P.C.); xtaylor@iu.edu (X.T.); 2Department of Anatomy, Cell Biology & Physiology, Indiana University School of Medicine, Indianapolis, IN 46202, USA

**Keywords:** cerebral amyloid angiopathy, tau, neuroinflammation, amyloid

## Abstract

Cerebral amyloid angiopathy (CAA) is typified by the cerebrovascular deposition of amyloid. Currently, there is no clear understanding of the mechanisms underlying the contribution of CAA to neurodegeneration. Despite the fact that CAA is highly associated with the accumulation of Aβ, other types of amyloids have been shown to associate with the vasculature. Interestingly, in many cases, vascular amyloidosis has been associated with an active immune response and perivascular deposition of hyperphosphorylated tau. Despite the fact that in Alzheimer’s disease (AD) a major focus of research has been the understanding of the connection between parenchymal amyloid plaques, tau aggregates in the form of neurofibrillary tangles (NFTs), and immune activation, the contribution of tau and neuroinflammation to neurodegeneration associated with CAA remains understudied. In this review, we discussed the existing evidence regarding the amyloid diversity in CAA and its relation to tau pathology and immune response, as well as the possible contribution of molecular and cellular mechanisms, previously associated with parenchymal amyloid in AD and AD-related dementias, to the pathogenesis of CAA. The detailed understanding of the “amyloid-tau-neuroinflammation” axis in the context of CAA could open the opportunity to develop therapeutic interventions for dementias associated with CAA that are currently being proposed for AD and AD-related dementias.

## 1. Introduction

Alzheimer’s disease (AD), the most common form of dementia, is characterized by the extracellular deposition of parenchymal β-amyloid (Aβ), intracellular accumulation of tau as neurofibrillary tangles (NFTs), synaptic loss, and significant inflammation [1,2]. Cerebral amyloid angiopathy (CAA) is typified by the cerebrovascular deposition of amyloid and has a close molecular relationship with AD, but remains clinically distinct. Vascular amyloid accumulation is identified in an estimated 85%–95% of individuals with AD [3,4], positioning CAA as one of the strongest vascular contributors to age-related cognitive decline [5,6]. Despite this high prevalence, the severity of CAA is highly variable in AD and therefore does not seem to strongly depend on the severity of AD pathology (e.g., CERAD and Braak stages) [7,8,9]. The mechanisms responsible for CAA pathogenesis and its downstream effects on the brain are complex and not completely understood. As for the origin of amyloid in CAA, the smooth muscle cells were originally proposed as the source of cerebral amyloid. Nevertheless, the sole contribution of smooth muscle cells to CAA is made less likely by the existence of amyloid deposits in capillaries in CAA patients. Based on this, it has been proposed that the amyloid is indeed derived from neurons and is drained along the perivascular interstitial fluid pathway of the brain parenchyma and leptomeninges, depositing along the vessels [10,11]. This perivascular drainage impairment sets in motion a self-reinforcing pathway by which worsening vascular amyloid accumulation leads to the activation of vascular injury pathways. This results in the impairment of vascular physiology, leading to a further increase in amyloid accumulation [12,13]. The damaged endothelium can produce proinflammatory cytokines that magnify neuroinflammation, glial activation, and secondary injury, leading to blood–brain barrier (BBB) breakdown [14]. Despite the fact that CAA is highly associated with the accumulation of Aβ [4], other types of amyloids have been shown to associate with the vasculature, suggesting that CAA should be considered as a group of biochemically and genetically diverse disorders unified by the accumulation of amyloid deposits in the walls of arterial blood vessels, and, in some cases, also in capillaries of the CNS parenchyma and leptomeninges [15,16,17]. Interestingly, in many cases, vascular amyloidosis is associated with severe neurofibrillary tau pathology, which is more pronounced around amyloid-laden blood vessels [18,19,20].

Over the past decades, a major focus of research has been on understanding the molecular and cellular connections between parenchymal amyloid, tau aggregation, and neuroinflammation, while the correlation between vascular amyloid pathology with tau aggregation and neuroinflammation has mainly been reported based on neuropathological observations.

In this review, we will summarize the existing evidence regarding the amyloid diversity in CAA and its relation to tau pathology and neuroinflammation, as well as discuss the possible contribution of molecular and cellular mechanisms, previously associated with parenchymal amyloid, to the pathogenesis of CAA.

## 2. The Diversity of Amyloid in CAA

Though CAA is defined by the accumulation of amyloid in the vasculature, distinct types of amyloids have been reported to deposit in blood vessels, as seen in Table 1. Broadly, there are Aβ-related CAAs and other amyloid-related CAAs [4,21,22,23]. In Aβ-CAA, the amyloidogenic forms of Aβ (Aβ_40_ and Aβ_42_) are produced from the processing of the 695 aa amyloid precursor protein (APP) by the sequential proteolytic cleavage of a β-secretase (BACE) and a γ-secretase-presinilin complex in what is called the amyloidogenic pathway, since it produces peptides that are prone to aggregate. This differs from the more physiological non-amyloidogenic pathway, where the sequential cleavage is produced by an α-secretase first and later by the γ-secretase-presinilin complex [24,25,26]. The transition from soluble to amyloidogenic nature of these Aβ peptides is explained by the conformational change given by the increase in the content of β-sheets in their structure, favoring their deposition as oligomers to finally form aggregated fibrils. The main type of Aβ accumulated in the vasculature is Aβ_40_ and to a lesser extent Aβ_42_ [3]. The vascular accumulation of amyloid species promotes the weakening of the blood vessel wall, challenging its integrity and leading to infarcts and hemorrhages [4,27]. This also causes smooth muscle degeneration and neurotoxicity [28].

### 2.1. Aβ Amyloid in CAA

Aβ-CAA can be divided into two main groups: a sporadic type (SCAA) and a hereditary type. SCAA constitutes the most common and the majority of the cases of Aβ-CAAs. In these, an age-related failure of elimination of amyloidogenic Aβ peptides is the most important risk factor for their accumulation in the vasculature [79]. On the contrary, in hereditary Aβ-CAAs, specific mutations are directly related with the vascular accumulation of Aβ. These are the cases of several rare familial genetic conditions termed hereditary hemorrhages with amyloidosis, caused by a mutation on the *APP* gene located on chromosome 21 [21,22,80,81]. The most studied of these is the Dutch type, where the substitution of glutamic acid for glutamine at codon 693 leads to the production of an aberrant Aβ_40_ that aggregates and accumulates rapidly in arterioles of meninges and brain cortex [29,30,31,32]. Three other less studied mutations of this same codon are the Italian, the Arctic, and the Osaka types. The first one presents a substitution of glutamic acid for lysine and the second for glycine, both upregulating an aberrant form of Aβ_40_ [33,34]. In the third one, the lack of a glutamate as a result of a whole deletion of codon 693 leads to the production of highly oligomeric Aβ_40_ and Aβ_42_ [35,36]. Mutations on other codons are reported to cause synthesis of abnormal forms of Aβ as well. An alanine replacement by glycine at codon 692 is present in the Flemish type. In this case, the change affects the cleavage site of the α-secretase on APP, shifting it towards β-secretase processing, upregulating both Aβ_40_ and Aβ_42_ [37]. On the other hand, in the Iowa type, a substitution of asparagine for aspartic acid at codon 694 causes an increase only of Aβ_40_ [38]. A substitution of leucine for valine at codon 705 is identified as the Piedmont variant, showing severe Aβ_40_ and Aβ_42_ vascular deposits [39]. Another Italian type was reported affecting codon 713, where the replacement of alanine for threonine causes extensive Aβ_40_ aggregation [40,41]. Additionally, codon 714 can be differently mutated giving rise to the Austrian and a rare Iranian type. In the first case, the mutation presents a change in a threonine for an isoleucine, directly affecting the γ-secretase cleavage site, increasing the Aβ_42_/Aβ_40_ ratio [42]. In the second case, the substitution is for alanine, and is likely to alter APP processing such that more Aβ_42_ is produced [43].

### 2.2. Non-Aβ Amyloid in CAA

Although the Aβ peptide is by far the most common amyloid accountable for the vascular accumulation and damage, other types of amyloid have been shown to cause the same effects in hereditary types of CAA [21,22]. This is the case of Familial British Dementia (FBD) and Familial Danish Dementia (FDD) [18,44,48]. Both conditions show progressive loss of cognitive functions, dementia and ataxia. Neuropathologically, FDD closely resembles FBD regarding its vascular amyloidosis; however, parenchymal deposits found in the hippocampus of patients with FDD were Congo red and Thioflavine-S (ThioS)-negative [19]. Interestingly, brains from affected individuals present tau aggregation [45]. Common to both FBD and FDD is the involvement of mutations on the *BRI_2_* gene on chromosome 13 that encodes the membrane-bound 266 aa BRI_2_ protein. Its physiological cleavage by protein convertases generates the soluble 23 aa BRI_2_-23 peptide [18,44,46]. However, two different mutations on the *BRI2* gene will lead to the production of a mutated form of BRI_2_. In individuals affected by FBD, a point mutation eliminates the normal stop codon on the gene. In FDD, individuals show a 10-nucleotide duplication causing a frameshift. In both cases, there is a read extension and a subsequent addition of 11 aa. The processing of these abnormal 277 aa mutated Bri_2_ proteins produce the 34 aa ABri and ADan amyloids in FBD and FDD respectively, both of which are highly amyloidogenic and neurotoxic [22,46,47,49,82]. In addition, it has been reported that ADan can be deposited in combination with Aβ in blood vessels of the brain parenchyma [44]. In the case of cerebral hemorrhage with amyloidosis of the Islandic type (HCHWA-I), the nature of extensive deposits of amyloid fibrils is the accumulation of the mutated form of the cystatin C transmembrane protein [51,52]. The mutation corresponds to a single nucleotide substitution at codon 68 of the *CST3* gene on chromosome 20 [51,83]. Remarkably, immunohistochemical analysis of brains of patients with AD revealed that this mutated peptide colocalizes with Aβ in parenchymal and vascular amyloid deposits [53]. Other mutated proteins that can accumulate in an amyloidogenic fashion are transthyretins (TTRs), gelsolin, and the prion protein (PrP) [21,22]. Several different mutations, see Table 1, on the *TTR* gene on chromosome 18 produce an amyloidogenic form of the TTR protein that aggregates extensively in leptomeninges [58]. This condition is termed meningovascular amyloidosis [22,80]. Interestingly, one of the cases analyzed showed distinctive aggregates of phospho-tau subjacent to TTR amyloid deposits in all regions of the neocortex and primary motor and striate cortices, indicating a potential link between TTR amyloid and cortical tauopathy [61]. In hereditary gelsolin amyloidosis, also called familial amyloidosis Finnish type (FAF), a mutation in the *GSN* gene on chromosome 9 causes gelsolin to be abnormally cleaved, generating several small fragments with amyloidogenic properties [75,76,77]. Finally, the amyloidogenic variant of the PrP (PrPSc) is caused by a single point mutation at codon 145 in the *PRNP* gene on chromosome 20, resulting in a premature stop codon that alters its glycosylation and signal sequence sites [20], leading to an increase of β-sheet structures in the protein [84]. These characteristics allows PrPSc to increase its propensity to aggregate in the vasculature [85]. A mutation at codon 163 has also been reported to produce a PrPSc [22,78]. Overall, these observations suggest that CAA is a heterogeneous group of CNS disorders, characterized by the dynamic accumulation of different amyloid species in the vasculature.

## 3. Perivascular Tau Aggregation and Its Interplay with Cerebrovascular Damage and CAA

Pathological aggregation of the microtubule-associated protein tau and the preponderance of NFTs or other inclusions containing tau are defining histopathological features of AD and many neurodegenerative diseases collectively known as tauopathies [86]. Interestingly, accumulation of tau has been detected in cerebrovascular pathologies associated with endothelial dysfunction and cognitive impairment [87,88]. Noteworthy, several studies have suggested that pathological changes of tau in neurons can impact brain endothelial cell biology, altering the integrity of the brain’s microvasculature [89,90]. For instance, it has been shown that vessel wall remodeling of leptomeningeal arteries is an early-onset, tau pathology-dependent process, which may potentially contribute to downstream CAA-dependent microvascular pathology in AD patients [90]. Even more, other studies have demonstrated an increase of BBB permeability and the accumulation of tau oligomers in the cerebral microvasculature of human patients with progressive supranuclear palsy (PSP) [91,92], emphasizing the role of tau aggregates in the functional and structural integrity of the cerebral vasculature. In a similar fashion, tau-overexpressing mice develop changes to blood vessels including abnormal, spiraling morphologies, reduced blood vessel diameter, and increased overall blood vessel density in the cortex [89]. In a different mouse model for tauopathies, BBB dysfunction emerges at the same time that perivascular tau emerges around major hippocampal blood vessels. However, when tau expression is suppressed, BBB integrity is preserved, suggesting that the BBB can be stabilized in a tauopathic brain by reducing tau levels [93]. Overall, these studies suggest a strong relation between tau pathology and vascular damage, and how tau aggregation is not a unidirectional event where vascular damage initiates a series of events that triggers tau aggregation, but is rather a vicious cycle where tau pathology enhances vascular damage.

In many cases, vascular amyloidosis is accompanied by significant perivascular tau pathology [19,20,94], supporting a unifying pathological mechanism in which vascular accumulation of amyloidogenic peptides triggers a complex pathological cascade leading to tau accumulation and neurodegeneration. However, it remains to be determined whether abnormal tau phosphorylation is the consequence of CAA or an independent disease characteristic. Recently, we biochemically demonstrated an increase of hyperphosphorylated forms of tau in the Tg-FDD mouse model characterized by vascular deposition of ADan amyloid [95]. This increase was only observed in certain phospho-tau epitopes, but not others that have been associated with tau NFTs in AD, suggesting that the process of tau hyperphosphorylation associated with CAA deposits could be different from the hyperphosphorylation of tau associated with parenchymal deposition of amyloid [96]. In the same study, we observed how these perivascular tau aggregates in the vicinity of vascular amyloid were associated with activated astrocytes [95]. Interestingly, astrocytes play a key role in maintaining the BBB via astrocytic endfeet directly opposed to vascular endothelial cells [96], and tau has been shown to accumulate in these endfeet in tauopathies [97,98], including perivascular astrocytic tau deposits in CAA patients [19]. Even more, the presence of perivascular tau aggregates has been reported in patients with chronic trauma encephalopathy (CTE) [99,100]. These tau deposits in CTE were also associated with activated astrocytes [99]. These results suggest that vascular damage, independent of its cause (amyloid accumulation or CTE), could induce an astrocytic response that triggers tau aggregation.

Previous studies have shown how endogenous WT tau appears to be required for parenchymal Aβ-amyloid accumulation and ApoE4 to cause synaptic, network, and cognitive deficits in mouse models of AD [101,102,103]. Significantly, the removal of endogenous tau expression in a mouse model for Parkinson’s disease, characterized by the aggregation of mutant alpha-synuclein, completely ameliorates cognitive dysfunction and concurrent synaptic deficits without affecting alpha-synuclein expression or accumulation of selected toxic alpha-synuclein oligomers [104]. Tau ablation has also been shown to attenuate motor abnormalities in a Huntington’s disease (HD) mouse model [105] and prevent deficits in spatial learning and memory after repeated mild frontal impact in WT mice [106]. Furthermore, tau reduction has the ability to block epileptogenesis of diverse causes, including epileptic activity triggered by pharmacological blockade of GABAA channels [104,107], genetic ablation of the voltage-gated potassium channel subunit Kv1.1 [108], depletion of ethanolamine kinase or of the K^+^–Cl^−^ cotransporter [108], or depletion of the voltage-gated sodium channel subunit Nav1.1 [109]. The mechanisms underlying these beneficial effects of tau reduction remain to be determined [86]. New evidence suggests that tau could be involved in a common pathway for neurodegeneration triggered by cerebrovascular abnormalities and parenchymal amyloid pathologies [88]. Recently, a novel study showed how exposing mice to a salt-rich diet not only leads to cognitive dysfunction associated with a nitric oxide deficit in cerebral endothelial cells and cerebral hypoperfusion, but also induces hyperphosphorylation of tau [110]. Remarkably, the authors did not observe salt-induced cognitive impairment in tau-null mice or in mice treated with anti-tau antibodies, despite the persistent cerebral hypoperfusion and neurovascular dysfunction [110].

Thus, considering the dependency of parenchymal amyloid for tau to exert neurotoxicity and the relevance of tau in several pathogenesis associated with neurovascular dysfunction, it is feasible to suggest that vascular amyloid would also depend on tau to trigger neurodegeneration and that partial tau reduction could be considered a feasible approach for the treatment of dementias associated with CAA.

## 4. The Implication of AD Immune-Risk Factors and Glial Response in CAA

The aggregation and accumulation of vascular amyloid has long been recognized as a major contributor to neuroinflammation, playing a fundamental role in the pathogenesis of AD. However, the majority of studies addressing this issue have focused on parenchymal amyloid deposition with little attention given to CAA [111,112,113,114]. Data from human patients and in vivo models for CAA suggest that damage to vessel walls activates the endothelium, facilitates the infiltration of monocytes/macrophages and incites glial reactivity, increasing the production of pro-inflammatory cytokines [115,116,117,118,119]. Novel genetic studies suggest that glial reactivity and innate immune cell activation may drive AD pathogenesis through a vascular-related mechanism [120]. Given the relationship between neuroinflammation and vascular damage, it is imperative to understand in detail the role of neuroinflammatory pathways associated with CAA pathogenesis.

The neurovascular unit (NVU) is a complex structure organized by endothelial cells, intimately associated with pericytes and astrocytic endfeet to form the BBB. Parenchymal cells including excitatory neurons, regulatory interneurons and microglia actively interact with the NVU [121]. As central components of the NVU, pericytes represent the first line of immunological defense as antigen-presenting cells, playing a pivotal role in the maintenance of the BBB and microvascular stability. They also participate in capillary constriction and dilation, regulating cerebral blood flow, immune cell entry, and clearance of macromolecules from the brain’s interstitial fluid [122,123]. When CAA occurs, pericytes display degenerative features influencing BBB breakdown, microaneurysms, neuroinflammation, and neurodegeneration [124,125,126,127]. When exposed to Aβ_40_, pericytes exhibit an increase in the activity of caspases-3 and -7, reducing their viability and promoting tau pathology that develops into early neuronal loss and cognitive changes [124]. Alongside pericytes, perivascular macrophages (PVM) are important immunoregulatory cells performing phagocytosis and responding to transient CNS inflammation [128]. PVM are distinguished from microglia by their expression of CD36, CD206, and acid phosphatase. They have a high rate of turnover and are involved in the movement of solutes, infectious agents and immune cells from the blood to the brain [129,130,131]. The depletion of PVM by directly injecting clodronate, a compound used to study innate responses to CNS injuries, into the left lateral ventricle of four-month-old TgCRND8 mice results in a significant increase in CAA in cortical and leptomeningeal blood vessels [132]. On the other hand, the sustained stimulation of PVM by injecting chitin, a long-chain polymer of N-Acetylglucosamine, caused a significant reduction of ThioS-labeled cortical blood vessels and CAA load [133,134]. It has been suggested that PVM requires the C-C chemokine receptor type 2 (CCR2), a receptor involved in the regulation of macrophage migration and infiltration, to remove amyloid from the brain, since APP^swe^ CCR2^−/−^ mice showed a drastic impairment in Aβ clearance and amplified CAA [135].

### 4.1. Microgliosis in CAA

Besides PVM, microglia are also important regulators of neuroinflammation, releasing a variety of proinflammatory and cytotoxic products in response to cerebral insult or injury. In AD patients and AD mouse models, microglia clusters around plaques and vascular amyloid, moving toward newly formed plaques within 24 h of their formation as well as toward existing plaques [136,137], displaying an activated phenotype capable of producing cytokines and chemokines such as IL-1, IL-6, TNF-*α*, TGF-*β*1, TGF-*β*2, MIP-1*α*, and MCP-1, enriching the brain’s neuroinflammatory profile [138]. Several studies have investigated the targeted elimination of microglial in 5xFAD mice through inhibition of colony stimulating factor 1 receptor (CSF1R), a microglial chemokine receptor, and showed that chronic microglial elimination at advanced stages of pathology does not alter amyloid-β levels or plaque load; however, it does rescue dendritic spine loss and prevent neuronal loss, as well as decrease overall neuroinflammation [139]. Surprisingly, elimination of microglia in the initial stages of the disease, during the plaque-forming period, appears to play a role in the pathogenesis of CAA as the sustained elimination of microglia parallels the loss of parenchymal plaque deposition and results in an immense shift in CAA pathology suggesting one function of microglia in the normal brain is to protect from CAA [140]. Numerous studies have focused on targeting plaque elimination through passive immunization and opsonization by microglia/macrophages through the peripheral administration of anti-Aβ antibodies. These studies have shown potential beneficial effects on plaque clearance and cognitive improvement; however, there are also toxic effects such as CAA-associated hemorrhage [141,142,143,144,145,146,147,148,149]. Additionally, the targeting of certain scavenger receptors expressed in innate immune cells and cerebral blood vessels has shown to have therapeutic effects for CAA. This is the case of CD36, a critical innate immune receptor present in endothelial cells and microglia/macrophages, involved in a key signaling pathway through which Aβ exerts its deleterious vascular effects through the production of reactive oxygen species. APP^swe^ mice with a genetic deletion of CD36 showed reduced CAA burden, improved neurovascular function, reduced smooth muscle cell fragmentation, a selective reduction in Aβ_40_ plaque load leaving Aβ_42_ amyloid levels unchanged, and improved cognitive performance, suggesting improved cerebrovascular health is able to preserve cognitive function irrespective of parenchymal amyloid burden [150].

### 4.2. Astrogliosis in CAA

At the same time, astrocytes also have the ability to respond to and influence immune and inflammatory responses following insults such as excitotoxicity, ischemia, apoptosis, necrosis, and inflammatory cues. This is achieved by undergoing a pronounced transformative state called ‘reactive astrogliosis’, regulating inflammatory responses that can be either neuroprotective or neurotoxic and participating in migratory, phagocytic, and proteolytic activity [151,152,153,154,155,156]. The interplay between apolipoprotein E (ApoE), Aβ, and astrocytes has gained important attention as ApoE, the most abundant apolipoprotein in the brain, is produced by astrocytes and has been widely confirmed to influence lipid metabolism and the processing, accumulation, and clearance of Aβ. Additionally, the ApoE4 allele mutation is the strongest genetic risk factor for both AD and CAA [157,158,159]. ApoE-deficient astrocytes lose their ability to internalize and degrade deposited Aβ peptides [160], with evidence suggesting that Aβ_40_ is internalized in vivo preferentially by astrocytes, not microglia [161,162]. The role of ApoE in CAA pathogenesis has gained important attention as it was observed that APP^swe^ mice expressing human ApoE4 show substantial CAA pathology, with very little parenchymal plaque deposition, as well as an early increase in the Aβ40/42 ratio, demonstrating that once Aβ fibrillogenesis occurs, ApoE4 expression results in a shift in the amyloid deposition from parenchyma to the vasculature [163]. In addition to ApoE, immunohistochemistry from human AD cases has also revealed a relationship of amyloid with apolipoprotein J or clusterin (ApoJ or CLU) and apolipoprotein A-I (ApoA-I), both with implications in CAA pathology [164]. CLU is the second major apolipoprotein produced by astrocytes, having roles in the aggregation, toxicity and BBB transport of Aβ, suggesting a strong role in CAA regulating the balance between Aβ deposition and clearance [165,166]. Additionally, loss of CLU in APP/PS1 mice results in a marked decrease in plaque deposition in the brain parenchyma and a striking increase in CAA, suggesting that the absence of CLU shifts Aβ to perivascular drainage pathways, resulting in fewer parenchymal plaques. To this extent, it was observed that adding CLU exogenously reduces Aβ_40_ and Aβ_42_ interactions with cerebral vessels [166]. ApoA-I has also been linked to AD pathology, binding to circulating Aβ peptides and occasionally associating with senile plaques [164,167]. Complete loss of ApoA-I increases cortical and hippocampal CAA pathology and astrogliosis in APP/PS1 mice, boosting both neuro and vascular inflammatory markers, specifically IL-1β, PDGFRβ, GFAP, and ICAM-1. These mice also showed an increase in GFAP-positive astrocytes associated with the cerebrovasculature [168]. ApoA-I could be an interesting target for a CAA-directed therapy as it was observed that APP/PS1 mice overexpressing ApoA-I or injection of ApoA-I reduced both astrogliosis and cerebral amyloid burden. Some groups have already begun to evaluate the potential benefits of ApoA-I mimetics in AD, observing benefits with respect to astrogliosis, amyloid pathology, CAA, and whole brain neuroinflammation [169,170,171]. Additionally to the aforementioned proteins, several AD genetic risk factors are immune-related genes [172,173,174]. Again, the presiding focus pertains to the relationship of these genes to parenchymal amyloid, with little consideration to vascular amyloid deposition. For instance, TREM2, one of the most studied AD-immune risk factors, has always been analyzed in the context of parenchymal amyloid [175,176,177,178,179]. This is also the case for genes such as *CR1*, *ABCA7*, *CD33*, *MEF2C*, *HLA-DRB1/DRB5*, *TRIP4*, *MS4A*, *EPHA1*, and *SPI1* [180,181]. Interestingly, recent genetic association studies of AD and other dementias have established correlations between specific risk factors such as *ABCA7*, *CR1*, *FERMT2*, *NME8*, *SLC24A4*, *SORL1*, *ZCWPW1*, and *GALNT7* to CAA, with *ABCA7* and *CR1* having the strongest association [182], enhancing the necessity of further cellular and molecular studies to dissect this relationship in detail.

## 5. Conclusions

Tau pathology and neuroinflammation have been identified as major contributors of neurodegeneration associated with amyloid deposits in AD and AD-related dementias. This contribution to amyloid pathology has been extensively studied in vivo utilizing animal models characterized by the accumulation of extra- or intracellular accumulation of parenchymal amyloid [103,104,179,183,184,185,186]. However, not many cellular and molecular studies have been performed to determine the role of tau and neuroinflammation in dementias associated with vascular amyloid accumulation, despite the fact that numerous pathological studies have suggested a preponderant neuroinflammatory response and perivascular tau aggregation in patients and in vivo models for CAA [19,50,95,118,187,188]. In this review, we discussed novel studies that link tau aggregation and neuroinflammation to neurovascular pathologies such as CAA and suggest how the “amyloid–tau–neuroinflammation” axis that has been studied in extensive detail in the context of parenchymal amyloid needs to be dissected in the context of vascular amyloid deposition, as seen in Figure 1. Further studies on the involvement of tau and neuroinflammation on dementias associated with CAA could open the opportunity to develop therapeutic interventions for CAA that are currently being proposed for AD and AD-related dementias [183,184,185], as seen in Figure 1.

## Figures and Tables

**Figure 1 ijms-20-06319-f001:**
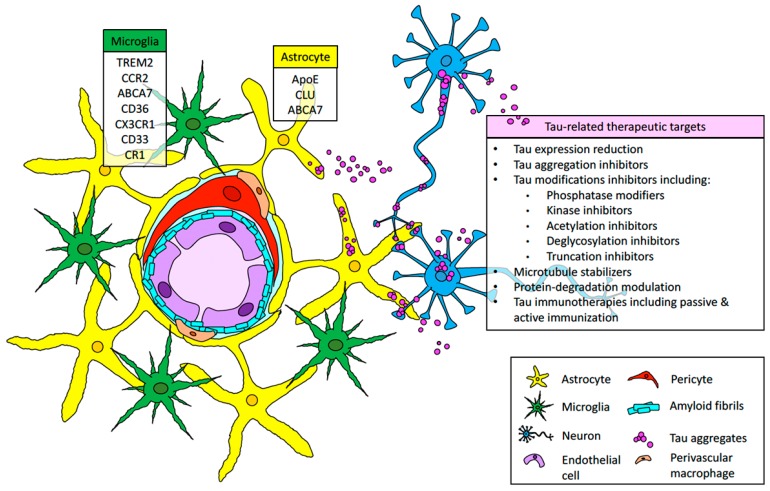
The amyloid-tau-neuroinflammation axis in the context of cerebral amyloid angiopathy (CAA). Amyloid accumulates in the vasculature, causing a destabilization of the neurovascular unit, translated in endothelial damage, pericyte decay, and perivascular macrophage, astrocyte, and microglial reaction. These series of events leads to neuroinflammation. Additionally, tau aggregates in astrocytes and neurons could contribute to neuroinflammation and neurodegeneration observed in CAA. A number of immune-related genes previously associated with AD and the inclusion of tau-related events could be signaled as possible therapeutic targets and a useful tool to contain the detrimental consequences of CAA.

**Table 1 ijms-20-06319-t001:** Sporadic and hereditary cerebral amyloid angiopathy (CAA) forms.

Amyloid	Gene	Precursor Protein	Mutation	Disease
Aβ	*APP*	Amyloid precursor protein	none	Sporadic CAA [21,22]
Aβ	*APP*	Amyloid precursor protein	E693Q	Hereditary Cerebral Hemorrhage with Amyloidosis Dutch type [29,30,31,32]
Aβ	*APP*	Amyloid precursor protein	E693K	Hereditary Cerebral Hemorrhage with Amyloidosis Italian type [33]
Aβ	*APP*	Amyloid precursor protein	E693G	Hereditary Cerebral Hemorrhage with Amyloidosis Arctic type [34]
Aβ	*APP*	Amyloid precursor protein	E693Δ	Hereditary Cerebral Hemorrhage with Amyloidosis Osaka type [35,36]
Aβ	*APP*	Amyloid precursor protein	A692G	Hereditary Cerebral Hemorrhage with Amyloidosis Flemish type [37]
Aβ	*APP*	Amyloid precursor protein	D694N	Hereditary Cerebral Hemorrhage with Amyloidosis Iowa type [38]
Aβ	*APP*	Amyloid precursor protein	L705V	Hereditary Cerebral Hemorrhage with Amyloidosis Piedmont type [39]
Aβ	*APP*	Amyloid precursor protein	A713T	Hereditary Cerebral Hemorrhage with Amyloidosis Italian type [40,41]
Aβ	*APP*	Amyloid precursor protein	T714I	Hereditary Cerebral Hemorrhage with Amyloidosis Austrian type [42]
Aβ	*APP*	Amyloid precursor protein	T714A	Hereditary Cerebral Hemorrhage with Amyloidosis Iranian type [43]
ABri	*BRI2*	British Amyloid protein	799A>T	Familial British Dementia [18,44,45,46,47]
ADan	*BRI2*	Danish Amyloid protein	787_796dupTTTAATTTGT	Familial Danish Dementia [18,45,46,48,49,50]
ACys	*CST3*	Cystatin C	L68Q	Hereditary Cerebral Hemorrhage with Amyloidosis Islandic type [51,52,53]
ATTR	*TTR*	Transthyretin	D18G	Meningovascular amyloidosis Hungarian variant [54,55,56,57]
ATTR	*TTR*	Transthyretin	V30G	Meningovascular amyloidosis Ohio variant [58,59,60]
ATTR	*TTR*	Transthyretin	Y69H	Meningovascular amyloidosis rare variant [61,62,63,64]
ATTR	*TTR*	Transthyretin	A25T	Meningovascular amyloidosis rare variant [65,66]
ATTR	*TTR*	Transthyretin	V30M	Meningovascular amyloidosis rare variant [67,68]
ATTR	*TTR*	Transthyretin	T49P	Meningovascular amyloidosis rare variant [69]
ATTR	*TTR*	Transthyretin	L58R	Meningovascular amyloidosis rare variant [70]
ATTR	*TTR*	Transthyretin	F64S	Meningovascular amyloidosis rare variant [71]
ATTR	*TTR*	Transthyretin	Y114C	Meningovascular amyloidosis rare variant [72]
ATTR	*TTR*	Transthyretin	L12P	Meningovascular amyloidosis rare variant [73]
ATTR	*TTR*	Transthyretin	G53R	Meningovascular amyloidosis rare variant [74]
AGel	*GSN*	Gelsolin	D187N or D187Y	Hereditary gelsolin amyloidosis or familial amyloidosis Finnish type [75,76,77]
PrPSc	*PRNP*	Prion protein	Y145stop	Gerstmann–Sträussler–Scheinker syndrome variant [20]
PrPSc	*PRNP*	Prion protein	Y163stop	Gerstmann–Sträussler–Scheinker syndrome variant [78]

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
