# Peer review of "The Amyloid-Tau-Neuroinflammation Axis in the Context of Cerebral Amyloid Angiopathy"

_ijms, 2019, doi:10.3390/ijms20246319_

Round 1

Reviewer 1 Report

The manuscript authored by Cisternas P. et al. is an interesting review about the Amyloid-Tau-Neuroinflammation axis in Cerebral Amyloid Angiopathy, in relation with beta-amyloid diversity. 

Despite the manuscript is well-written and balanced, here below my concerns and suggestions:

1) carefully revise the manuscript concearning English editing. In details, at lines 45-49 (rephrase 47-48). Sobstitute "leading to further increased amyloid accumulation" with "leading to a further increase in amyloid accumulation". Revise lines 266,287 and 320. 

2) The titles of the chapters are to a broader extent, thus in turn modify them being more specific.

3) When possible include subparagraph for each chapter, thus to be more reader friendly. 

4) Revise Fig.1 not using the same colour for Amyloif fibrils and microglia thus to be more evident and not to leave empty the inner of the vasculature not to mislead the reader. 

Reviewer 2 Report

I did not find any issues in this paper; I can only send congratulations for the Authors and recommend this manuscript for publication in IJMS.
